# EXPECTATION CURVATURE: BEYOND THE HESSIAN IN NON-SMOOTH LOSS LANDSCAPES

## ABSTRACT

Second-order methods seek to exploit loss curvature, but in deep networks the Hessian often fails to approximate it well, especially near sharp gradient transitions induced by common activation functions. We introduce an analytic framework that characterizes *curvature of expectation*, showing how such transitions generate pseudorandom gradient perturbations that combine into a glass-like structure, analogous to amorphous solids. From this perspective we derive: (i) the density of gradient variations and bounds on expected loss changes, (ii) optimal kernels and sampling schemes to estimate both Hessian and glass curvature from ordinary gradients, and (iii) quasi-Newton updates that unify these curvature terms with exactness conditions under Nesterov acceleration. To probe their empirical role, we implement ALICE, a lightweight diagnostic that inserts curvature estimates into controlled updates, revealing which terms genuinely influence optimization. In this way, our results support further optimization research: they introduce a new theoretical picture of nonsmooth loss landscapes that can catalyze future advances in pruning, quantization, and curvature-aware training.

## 1 INTRODUCTION

First-order optimizers such as SGD and Adam remain standard practice in deep learning, yet their efficiency is limited by the lack of curvature information. Second-order methods promise faster convergence by incorporating local curvature through the Hessian (Becker et al., 1988; LeCun et al., 1989; Bottou et al., 2018). However, deep networks with rectified linear units (ReLUs) and related activations introduce steep gradient transitions that break the smoothness assumptions underpinning Hessian-based methods. In these regimes, instantaneous second derivatives are poorly suited for extrapolation, leading to unreliable curvature estimates.

Across many parameters, each activation defines a boundary in parameter space. Crossing such a boundary changes gradient flow abruptly, producing pseudorandom perturbations that accumulate throughout the network. Collectively, these boundaries form what we call a *gradient glass*: a dense field of small, pseudorandom gradient jumps induced by ReLU activation boundaries. A gradient glass behaves analogously to amorphous solids: locally coherent within small domains but globally disordered. This structure accumulates from the many activation boundaries encountered along typical parameter updates. Figure 1 illustrates this idea in two dimensions: gray lines mark activation boundaries, and blue arrows indicate pseudorandom gradient shifts encountered upon crossing them. A hypothetical training trajectory (violet) passes through successive domains. The right panel shows how gradient and loss vary along the trajectory, with expectation curves providing consistent bounds on the range of fluctuations. We quantify these effects formally in Theorem 1 and Theorem 4. Parameters followed by ReLUs are influenced by glass effects because small perturbations can cross downstream activation boundaries, whereas parameters not followed by ReLUs (e.g., the final block) exhibit smoother, more Hessian-like behavior. Figure 2 empirically demonstrates this phenomenon.

**An Empirical Motivation** To test whether Hessian curvature explains these effects, we measured how gradient variations scale with perturbation distance. Let $\mathcal{L}(\boldsymbol{\theta})$ denote average loss over training batches with gradient $g(\boldsymbol{\theta})$ for parameters $\boldsymbol{\theta} \in \mathbb{R}^d$. Perturbing parameters by a random Rademacher

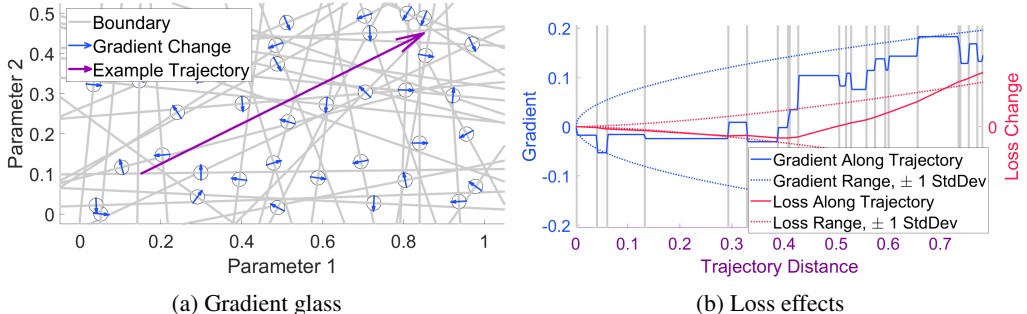

(a) Gradient glass          (b) Loss effects

Figure 1: Left: Illustration of gradient glass in a simplified 2D model. See Appendix B for details. Gray lines are domain boundaries due to changing a ReLU state. Blue arrows show pseudorandom gradient perturbations. Right: Moving along the example trajectory on the left (violet arrow), we track the cumulative gradient in the direction of displacement and integrate the corresponding loss. The gray bands show encountered gradient discontinuities. The dotted curves are expectation bounds derived in Sec. 3, showing that the realized gradients and loss are constant with these bounds.

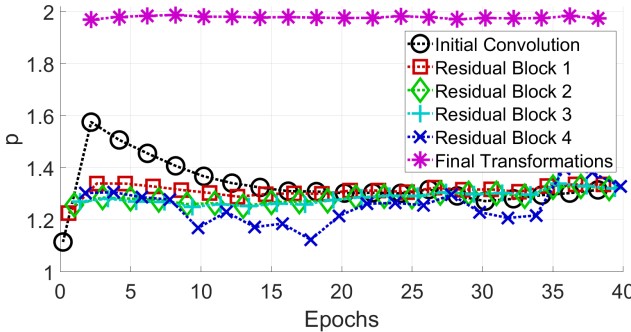

Figure 2: Scaling of gradient variations with perturbation distance. A pure Hessian contribution gives $p = 2$. Observed exponents $p < 2$ indicate additional curvature effects beyond the Hessian.

vector $\boldsymbol{\delta}$ of fixed distance $\lambda$, we measure gradient variations [1]

$$\boldsymbol{v}(\lambda) = \mathbb{E}_{\boldsymbol{\delta}}\left[\boldsymbol{\gamma}^2\right] = k\lambda^p \quad \text{where} \quad \boldsymbol{\gamma} = g(\boldsymbol{\mu} + \lambda\boldsymbol{\delta}) - g(\boldsymbol{\mu}). \tag{1}$$

Here and going forward, all vector powers are taken elementwise. If variations were dominated by the Hessian, we would expect quadratic scaling ($p = 2$). Yet experiments with ResNet18 on CIFAR-10 reveal consistently smaller exponents (Figure 2), indicating subquadratic growth. To estimate the local exponent $p$, we use a paired finite-difference procedure directly along the training trajectory. For each mini-batch, direction $\boldsymbol{\delta}$, and current parameter $\boldsymbol{\mu}$, we evaluate $v_1 = \left(g(\boldsymbol{\mu} + \lambda\boldsymbol{\delta}) - g(\boldsymbol{\mu})\right)^2$ and $v_2 = \left(g(\boldsymbol{\mu} + 2\lambda\boldsymbol{\delta}) - g(\boldsymbol{\mu})\right)^2$ using the same batch, direction, and perturbation center. Because $\mathbb{E}[v_i] \approx k(i\lambda)^p$, the local exponent is given by $p \approx \log_2(v_2) - \log_2(v_1)$. We maintain running averages of $v_1$ and $v_2$ using the $\beta_2$ exponential moving average (as in Adam), aggregate the resulting exponent across parameter groups, and plot its evolution over training. This measures the smoothness exponent in the immediate vicinity of the optimization path. No low-order Taylor expansion can yield $p < 2$, suggesting that gradient discontinuities contribute an additional source of curvature beyond the Hessian. We further observe that $p = 2$ dependence is recovered for parameters that follow the final activation layers, underscoring the connection between curvature behavior and activation-induced gradient discontinuities.

**Contributions** We present an analytic framework for *curvature of expectation*, capturing how loss evolves under parameter perturbations in the presence of activation-induced gradient discontinuities. Our main contributions are:

---

[1]The Rademacher vector contains i.i.d. coordinates with equal probability of taking $\pm 1$, which yields the minimal-variance diagonal estimator in Theorems 2 and 3.

1. a theoretical model of *gradient glass*, quantifying the density of gradient variations induced by ReLU-like activation boundaries,

2. derivation of an optimal kernel for unbiased, minimum-variance diagonal estimation from randomized matrix–vector products,

3. identification of the optimal perturbation distribution, showing that Rademacher samples minimize estimator variance and recover Hutchinson trace estimation as a special case,

4. analytic bounds on expected loss changes, yielding a $3/2$ power-law dependence that complements Hessian-based quadratic growth,

5. a modified quasi-Newton update that unifies Hessian and glass curvature terms, producing per-coordinate effective curvature with provable stability properties, and

6. exactness conditions under Nesterov acceleration, linking momentum and damping parameters to guarantee correction of hidden linear gradient components.

Together, these results establish a principled theory of curvature in nonsmooth networks, explain long-standing discrepancies in Hessian-based methods, and provide concrete algorithms for making this hidden structure measurable and actionable.

**Alice**   To ground the theory in experiment, we introduce ALICE, a streamlined probe designed to expose the role of curvature extraction. Alice functions as an investigative instrument: it isolates Hessian and glass contributions through controlled updates, providing clear evidence of when curvature-of-expectation matters. Our experimental results with Alice show that curvature terms are not only theoretically coherent, but empirically accessible, opening paths for future work in pruning, quantization, regularization, and the design of curvature-aware training methods.

## 2   RELATED WORK

**Second-Order Optimization**   Second-order methods approximate curvature to accelerate training beyond first-order updates. Early work incorporated Hessian information into multilayer perceptrons, computing diagonal terms for pruning (LeCun et al., 1989) and improving convergence through quasi-Newton (QN) updates (Becker et al., 1988). A comprehensive survey is provided by Bottou et al. (2018). Recent techniques avoid explicit Hessian construction by using matrix-vector products, enabling scalable spectral analysis and visualization of loss landscapes (Yao et al., 2020).

**Active Developments**   Curvature remains a focus of active research due to its link with generalization. Plasticity and catastrophic forgetting have been tied to Hessian eigenvalues (Lyle et al., 2023; Kong et al., 2023). Other studies connect curvature to transfer performance (Hemati et al., 2023), adversarial robustness (Li and Spratling, 2023), and federated optimization (Sen et al., 2023). Sharpness-Aware Minimization (SAM) leverages Hessian eigenvalue control to improve flatness (Kaur et al., 2023), while suppression of curvature along training trajectories can guide convergence to flat minima (Lee et al., 2023a;b). Extensions combining QN methods with Nesterov acceleration (Ninomiya, 2017; Indrapriyadarsini et al., 2020) and variational inference (Duersch, 2024) further illustrate the breadth of ongoing developments.

**Hessian Shortcomings and Glass Analogies**   Despite these advances, Hessian-based curvature is not reliable near sharp activation transitions. For ReLU-like units, extrapolation based on instantaneous second derivatives clearly cannot anticipate the gradient discontinuities (Figure 3). Empirical studies confirm the jagged loss topography that emerges in quantization (Frumkin et al., 2023) and the $v$-shaped loss structures observed along random directions (Li et al., 2024). These observations align with results from statistical physics, where spin-glass analogies explain the proliferation of saddle points and the role of curvature rectification (Dauphin et al., 2014; Parisi, 2007). Later work connects glass-like behavior to generalization phase transitions (Choromanska et al., 2015; Spigler et al., 2019). Our approach builds on this perspective by modeling gradient discontinuities as a *gradient glass*, and by deriving analytic tools to exploit the resulting curvature of expectation for optimization.

Recent analyses continue to examine geometric structure in loss landscapes, including modern characterizations of curvature and sharpness (Dauphin et al., 2024; Foret et al., 2021). These approaches focus primarily on smooth curvature—eigenvalues, trace properties, and sharpness-aware

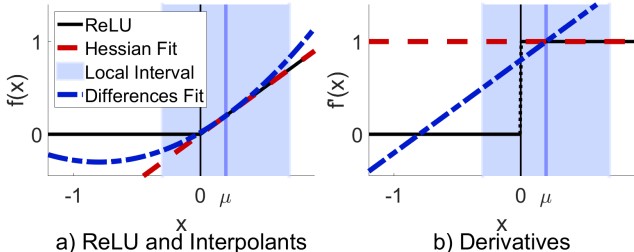

a) ReLU and Interpolants          b) Derivatives

Figure 3: ReLU extrapolation. The blue curve is the quadratic fit obtained from finite-difference estimates of the gradient over the local interval. For smooth functions this corresponds to a locally averaged Hessian. Here, it illustrates a better quadratic approximation around the ReLU kink at 0. Pure Hessian fitting cannot anticipate gradient discontinuities.

regularization—whereas our framework isolates curvature contributions that arise specifically from activation-induced gradient discontinuities. The two views are complementary: SAM and related methods regulate smooth curvature, while curvature-of-expectation characterizes the nonsmooth component that persists even when Hessian eigenvalues are well-behaved.

## 3 EXTRACTING AND EXPLOITING GLASS-LIKE CURVATURE OF EXPECTATION

We now present our framework to model curvature in networks with gradient discontinuities induced by ReLUs. Each ReLU introduces a parameter boundary where the gradient shifts abruptly, and collectively these create a glass-like structure in the loss landscape. We quantify this effect by deriving the density of gradient variations and show how it complements Hessian curvature. Using randomized gradient evaluations, we obtain optimal estimators for both glass and Hessian diagonals, derive bounds on expected loss changes, and construct a modified quasi-Newton step. We also establish exactness conditions for Nesterov acceleration under this structure. These results provide the foundation for curvature-aware optimization in nonsmooth settings. All proofs appear in Appendix A.

**Theorem 1** (Glass from ReLUs). *Consider a network with a sufficiently large number of ReLUs so that, within any small perturbation window, the subset of near-threshold activations is large enough to be well-approximated by a smooth density. Hold network inputs fixed and assume each pre-activation is locally linear in small parameter perturbations $\boldsymbol{\delta}$ centered at $\boldsymbol{\mu}$:*

$$y(\boldsymbol{\mu} + \boldsymbol{\delta}) = y(\boldsymbol{\mu}) + \boldsymbol{\delta}^{\top}\hat{\boldsymbol{\gamma}} \quad with \quad \hat{\boldsymbol{\gamma}} = \nabla_{\boldsymbol{\mu}}y(\boldsymbol{\mu}). \tag{2}$$

*Here, $y(\boldsymbol{\mu})$ denotes the vector of pre-activations and $\boldsymbol{\mu} \in \mathbb{R}^d$ are all network parameters. Let the upper bound on pre-activation shifts be $|\boldsymbol{\delta}^{\top}\hat{\boldsymbol{\gamma}}| \leq \Delta$, and define the set of near-threshold units $\mathcal{S}_{\Delta} = \{k : |y^{(k)}(\boldsymbol{\mu})| < \Delta\}$ where $y^{(k)}$ is the preactivation of unit $k$, and $z^{(k)} = \max(0, y^{(k)})$ is its ReLU output. Within this set, suppose pre-activations are uniformly distributed in $[-\Delta, \Delta]$, a simplifying assumption to capture networks with pre-activation densities that do not change rapidly in the vicinity of small parameter perturbations.*

*If gradient jumps from individual activation flips are independent, zero-mean pseudorandom variables, then the vector of gradient-variations $\boldsymbol{v}(\boldsymbol{\delta})$ given by Equation* (1) *satisfies*

$$\boldsymbol{v}(\boldsymbol{\delta}) \leq \boldsymbol{R}|\boldsymbol{\delta}| \quad where \quad \boldsymbol{R}_{ij} = \tfrac{1}{2\Delta} \sum_{k \in \mathcal{S}_{\Delta}} \left(\frac{d\mathcal{L}(\boldsymbol{\mu})}{dz^{(k)}}\right)^2 \hat{\boldsymbol{\gamma}}_i^{(k)2} |\hat{\boldsymbol{\gamma}}_j^{(k)}|. \tag{3}$$

The density matrix $\boldsymbol{R}$ captures gradient variation per unit length, beyond the Hessian. Any activation with ReLU-like derivative discontinuities will induce such terms, and averaging over inputs diminishes but does not eliminate their effects. Pointwise Hessians cannot detect them. Because each activation boundary is most sensitive to perturbations aligned with its largest-magnitude pre-activation gradient coordinates, which also induce the largest gradient jumps, the largest coordinates of $\hat{\boldsymbol{\gamma}}^{(k)}$ push their mass disproportionately onto the diagonal as cubic terms, rather than quadratic-linear products. Summing over many such contributions amplifies this diagonal dominance, making a diagonal approximation of $\boldsymbol{R}$ a reasonable simplification in practice. For this reason, we restrict our estimator

to the diagonal, mirroring standard diagonal quasi-Newton. Recovering full off-diagonal structure would require many more matrix–vector evaluations and is not feasible at model scale.

**Theorem 2** (Optimal Kernel for Diagonal Estimation). *Let $M$ be a linear operator accessible through matrix–vector products $y = M\delta$. Let $p(\delta) = \prod_{i=1}^{d} p(\delta_i)$ be a product distribution with i.i.d. zero-mean, unit-variance coordinates. For each $i \in [d]$, define the relative off-diagonal mass $\omega_i^2 = \sum_{j \neq i} M_{ij}^2 / m_i^2$ where $m = \mathrm{diag}(M)$. A kernel $\kappa_i(\cdot)$ is a scalar function applied to the perturbation coordinate $\delta_i$ to construct a diagonal estimator $\kappa_i(\delta_i)\, y_i \approx m_i$, thereby generalizing Hutchinson-style estimation. Unbiasedness requires that*

$$\mathbb{E}[\kappa_i(\delta_i)\, y_i] = m_i. \tag{4}$$

*Among all such kernels, the unique minimum-variance choice is*

$$\kappa_i^*(\delta_i) = \frac{c^{-1}\,\delta_i}{\delta_i^2 + \omega_i^2}, \qquad c = \int \frac{\delta_i^2}{\delta_i^2 + \omega_i^2}\, dp(\delta_i). \tag{5}$$

This applies to any i.i.d. sample density. Optimizing the density itself simplifies the kernel further.

**Theorem 3** (Optimal Perturbation Density). *Among product perturbation densities with i.i.d. factors $p(\delta) = \prod_{i=1}^{d} p(\delta_i)$, each with zero mean and unit variance, the variance-minimizing choice in Theorem 2 is the Rademacher distribution on each factor, i.e. probability $1/2$ of $\pm 1$. Then Equation (5) simplifies to $\kappa^*(\delta_i) = \delta_i$.*

This result recovers Hutchinson-style trace estimation as a special case, but here derived formally to permit estimation of diagonals for both $R$ as well as the Hessian. Going forward, we denote the diagonal glass-density as $\rho = \mathrm{diag}(R)$.

**Theorem 4** (Curvature of Expectation in Glass Loss). *As we displace parameters by $\delta$, the increase in loss $\Delta\mathcal{L}(\delta) = \mathcal{L}(\mu + \delta) - \mathcal{L}(\mu)$ is maximized if we enforce a local floor $\Delta\mathcal{L}(\delta_k) \geq 0$ by folding symmetric gradient-jump effects to obtain a valid convex upper model for the expectation. Using the diagonal approximation $v \approx \rho \odot |\delta|$, where $\odot$ is the Hadamard (elementwise) product, with $\rho \succeq 0$, and treating coordinates independently, we integrate the cumulative contribution of expected gradient jumps encountered along the displacement. The resulting expected increase in loss is bounded by*

$$\mathbb{E}[\Delta\mathcal{L}(\delta)] \leq \sqrt{\tfrac{2}{3\pi}}\; \rho^{\frac{1}{2}T}|\delta|^{\frac{3}{2}}. \tag{6}$$

The floor ensures steps increase expected loss, providing a coordinate-wise majorizer for the expected loss change. This effect is consistent with the reflections documented by Li et al. (2024). Otherwise, the symmetry of gradient changes would yield a flat expectation. Theorem 5 combines this majorized glass term with the gradient and diagonal Hessian to yield the optimal quasi-Newton updates.

**Theorem 5** (Modified Quasi-Newton Step). *Combining the bound of Equation (6) with the gradient $g$ and nonnegative Hessian diagonal $h \succeq 0$ gives*

$$\mathcal{L}(\mu + \delta) \leq \mathcal{L}(\mu) + \delta^\top\Big(g + \tfrac{1}{2}h \odot \delta\Big) + \sqrt{\tfrac{2}{3\pi}\,\rho^\top|\delta|^3}. \tag{7}$$

*Equation (7) implies that each coordinate contributes an additive convex upper model consisting of a linear term (gradient), quadratic term (Hessian), and a $3/2$-power term (glass curvature). Minimizing this coordinate-wise surrogate yields a closed-form update that can be interpreted as a quasi-Newton step with curvature modified by the glass term. Specifically, $\delta = -g \odot \bar{h}^{-1}$ with the modified Hessian*

$$\bar{h} = \hat{h} + h + \sqrt{\hat{h} \odot (\hat{h} + 2h)} + \varepsilon, \quad \hat{h} = 3\,\rho \odot (4\pi|g| + \varepsilon)^{-1}, \tag{8}$$

*where $\varepsilon > 0$ ensures numerical stability.*

Here $\varepsilon > 0$ ensures numerical stability. The effective curvature interpolates between Hessian and glass terms, shortening optimal steps in both regimes.

Having established how glassy gradient fluctuations lead to a modified quasi-Newton step with diagonal correction (Theorem 5), a central question in curvature-aware optimization is how curvature corrections interact with momentum-based accelerations such as Nesterov's method. This question

---

**Algorithm 1** Alice Topography Update

---

**Input**: evaluation center: $\boldsymbol{\nu}$; gradient function: $g(\boldsymbol{\theta})$.
**Input and Output**: running averages: $\boldsymbol{g}$, $\boldsymbol{\rho}$, and $\boldsymbol{h}$.
**Hyperparameters**: $\lambda, \beta_1, \beta_2$.
  1: Draw Rademacher $\boldsymbol{t}$.
  2: Evaluate $\boldsymbol{g}^{(\pm)} = g(\boldsymbol{\nu} \pm \lambda\boldsymbol{t})$ and $\boldsymbol{g}^{(0)} = g(\boldsymbol{\theta})$.
  3: $\boldsymbol{g} \leftarrow \beta_1\boldsymbol{g} + (1-\beta_1)\boldsymbol{g}^{(0)}$.
  4: $\boldsymbol{h} \leftarrow \beta_2\boldsymbol{h} + (1-\beta_2)\frac{1}{2\lambda}\big|\boldsymbol{g}^{(+)} - \boldsymbol{g}^{(-)}\big|$.
  5: $\boldsymbol{\rho} \leftarrow \beta_2\boldsymbol{\rho} + (1-\beta_2)\frac{2}{\lambda}\Big(\frac{1}{2}(\boldsymbol{g}^{(+)} + \boldsymbol{g}^{(-)}) - \boldsymbol{g}^{(0)}\Big)^2$.

---

has been studied extensively in quasi-Newton acceleration, and here we derive the specific conditions under which momentum and curvature remain algebraically consistent. Nesterov's method separates the parameter update from the evaluation point, providing an opportunity to test whether the quadratic model with $\bar{h}$ can be made consistent with accelerated gradient dynamics. Theorem 6 shows that a specific choice of update and evaluation fractions aligns the quasi-Newton model with Nesterov acceleration, while ensuring contraction of residual gradient errors.

**Theorem 6** (Exact Nesterov Accelerated Quasi-Newton). *In Nesterov acceleration, parameters and gradient evaluations evolve by*

$$\boldsymbol{\mu}^{(s+1)} = \boldsymbol{\mu}^{(s)} + \varphi\,\boldsymbol{\delta}^{(s)}, \qquad\qquad \boldsymbol{\nu}^{(s+1)} = \boldsymbol{\mu}^{(s)} + \omega\,\boldsymbol{\delta}^{(s)}, \qquad (9)$$
$$\boldsymbol{g}^{(s+1)} = \beta_1\boldsymbol{g}^{(s)} + (1-\beta_1)g(\boldsymbol{\nu}^{(s)}), \qquad\qquad\qquad\qquad (10)$$

*where $\boldsymbol{\delta}^{(s)}$ is the tentative update, $\varphi \leq 1$ is the update fraction, and $\omega \geq \varphi$ is the evaluation fraction. Suppose we approximate the loss quadratically with the modified Hessian $\bar{h}$ from Equation* (8), *so that the local gradient is*

$$g(\boldsymbol{\mu}^{(s)} + \boldsymbol{\delta}) = \boldsymbol{g}^{(s)} + \bar{h} \odot \boldsymbol{\delta}, \qquad (11)$$

*with optimal step $\boldsymbol{\delta}^{(s)} = -\boldsymbol{g}^{(s)} \odot \bar{h}^{-1}$. Yet assume the true dependence is linear but unknown*

$$g^*(\boldsymbol{\mu}^{(s)} + \boldsymbol{\delta}) = g^*(\boldsymbol{\mu}^{(s)}) + \boldsymbol{H}\boldsymbol{\delta} \qquad (12)$$

*that remains valid through $\boldsymbol{\delta}^{(s)}$. Then choosing $\varphi = 1 - \beta_1$ and $\omega = 1$ ensures that the momentum update Equation* (10) *matches the quadratic model Equation* (11)*, while capturing the unknown linear dependence in Equation* (12)*. Moreover, if*

$$\boldsymbol{g}^{(s)} = g^*(\boldsymbol{\mu}^{(s)}) + \boldsymbol{\gamma}^{(s)} \quad \text{then} \quad \boldsymbol{g}^{(s+1)} = g^*(\boldsymbol{\mu}^{(s+1)}) + \beta_1\boldsymbol{\gamma}^{(s)}, \qquad (13)$$

*so errors are reduced by factor $\beta_1$ each step.*

This links the damping factor and momentum parameter, showing that $\beta_1$ governs memory, damping, and correction of off-diagonal effects simultaneously.

**Summary** Theorems 1–6 provide a compact pipeline: ReLU-induced gradient jumps produce a measurable glass density (Theorem 1); diagonal elements of both glass and Hessian operators can be estimated optimally from randomized matrix–vector products when using Rademacher samples (Theorem 2, 3); the glass term tightens expected loss growth to a $3/2$ power law (Theorem 4), which directly yields a modified per-coordinate curvature used in quasi-Newton updates (Theorem 5); finally, damping and Nesterov-style evaluation choices make these updates stable and exact for hidden linear corrections (Theorem 6). This framework applies to any activation whose gradient changes rapidly over small intervals, thus defeating pure Hessian-based extrapolation. Full proofs are in Appendix A.

## 4 EXPERIMENTS AND RESULTS

To assess our theoretical framework empirically, we implement *Alice*, a lightweight optimization testbed. Alice is intended to enable diagnostic probing of curvature effects during optimization

by implementing both Hessian- and glass-based curvature approximations using ordinary gradient evaluations and inserting them into controlled training updates. Batch averaging reduces noise in both gradient and Hessian estimates by the usual $1/\sqrt{B}$ factor. Because the glass-density estimator squares the symmetric finite-difference, its batch noise is suppressed as $1/B$. Perturbation radii are chosen within the scale of typical parameter updates. This allows us to measure which curvature terms matter for different architectures, test the predictions of Theorems 4–6, and situate glass curvature alongside Hessian curvature in practical settings.

## 4.1 ALICE AS A PROBE

By construction, Alice is minimal: it strips away auxiliary design choices in optimizers so as to isolate the effect of curvature terms. Its role is to provide a consistent baseline for comparing Hessian, glass, and combined curvature in practice. While its performance is competitive, this is incidental—the purpose is not to propose "yet another optimizer," but to demonstrate how curvature-of-expectation can be estimated and how it shapes optimization dynamics. In downstream work, more sophisticated methods may leverage these terms differently; Alice provides the measurement baseline against which such designs can be assessed.

**Empirical Questions.** Our experiments are organized to probe three aspects of the theory: (i) Do glass curvature terms improve the effectiveness of quasi-Newton updates in practice? (ii) Does Nesterov acceleration capture hidden linear gradient structure as predicted (Theorem 6)? (iii) How do combined Hessian + glass updates affect step size and stability (Theorem 5)? All results should be read in this diagnostic light. Although Alice is designed as a diagnostic tool rather than a production optimizer, evaluating validation behavior during training provides the clearest way to reveal which curvature components materially influence loss navigation.

**Estimating Curvature.** To compute the three parameter-length quantities $g$, $\rho$, and $h$, Alice uses three gradient evaluations anchored at $\nu$ to remain consistent with Theorem 6,

$$\boldsymbol{g}^{(\pm)} = g(\boldsymbol{\nu} \pm \lambda\boldsymbol{\delta}) = g(\boldsymbol{\nu}) \pm \lambda\boldsymbol{H}\boldsymbol{\delta} + \boldsymbol{\gamma}^{(\pm)} \quad \text{and} \quad \boldsymbol{g}^{(0)} = g(\boldsymbol{\nu}). \tag{14}$$

This gradient model captures the average linear dependencies as matrix-vector multiplies $\boldsymbol{H}\boldsymbol{\delta}$, while $\boldsymbol{\gamma}^{(\pm)}$ express random fluctuations due to glass density via

$$\boldsymbol{H}\boldsymbol{\delta} = \mathbb{E}\Big[\big(\boldsymbol{g}^{(+)} - \boldsymbol{g}^{(-)}\big)/(2\lambda)\Big], \tag{15}$$

$$\boldsymbol{R}|\boldsymbol{\delta}| = \mathbb{E}\Big[\Big(\tfrac{1}{2}(\boldsymbol{g}^{(+)} + \boldsymbol{g}^{(-)}) - \boldsymbol{g}^{(0)}\Big)^2/(2\lambda)\Big]. \tag{16}$$

Here, expanding Equation (14) shows that the symmetric finite difference isolates the jump components: $1/2(\boldsymbol{g}^{(+)} + \boldsymbol{g}^{(-)}) - \boldsymbol{g}^{(0)} = 1/2(\boldsymbol{\gamma}^{(+)} + \boldsymbol{\gamma}^{(-)})$. The average second moment, scaled by $1/(2\lambda)$, yields the local density of gradient variations. Algorithm 1 summarizes the running update. Further implementation details appear in Appendix B.

## 4.2 EXPERIMENTAL RESULTS

**Nesterov Acceleration.** We now test whether the exactness property of Theorem 6 is observed in practice. Figure 4 examines the effect of NAQ when applied with different curvature terms[2]. Our ViT model uses linear-complexity attention (Shen et al., 2021) trained on $64 \times 64$ downsampled Imagenet (Deng et al., 2009; Chrabaszcz et al., 2017). Results in Fig. 4 and Tables 1–2 confirm these expected improvements. For ResNet18 on CIFAR-10, Nesterov acceleration ($\varphi = 0.1$) consistently outperforms the unaccelerated case across ten random seeds, with the gradient-glass term $\rho$ yielding the best median accuracy and the combination $\rho + h_{\mathrm{abs}}$ achieving the best maximum accuracy. Although the RMS Hessian approximation trails slightly, its performance remains competitive. For the ViT on downsampled ImageNet, the same pattern holds: acceleration again provides a clear gain, and the combined $\rho + h_{\mathrm{abs}}$ term delivers the lowest minimum and median losses across five seeds.

---

[2]In Figures and Tables, $\boldsymbol{\rho}$ denotes glass-based curvature from Line 5 and $\boldsymbol{h}_{\mathrm{abs}}$ denotes the absolute Hessian from Line 4. We also show the combined curvature from Theorem 5, written as $\boldsymbol{\rho} + \boldsymbol{h}_{\mathrm{abs}}$, and an RMS variant, $\boldsymbol{h}_{\mathrm{rms}}$, similar to AdaHessian (Yao et al., 2021).

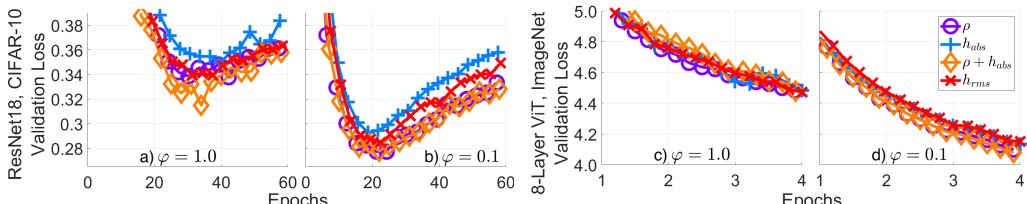

Figure 4: Validation loss over training epochs. Panels (a)–(d) compare full quasi-Newton updates ($\varphi = 1.0$) with NAQ updates ($\varphi = 0.1$). (a,b) ResNet-18 on CIFAR-10 (median over 10 seeds). (c,d) 8-layer ViT on ImageNet (median over 5 seeds). Each curve shows the median validation loss for the indicated curvature model[2]. ResNet18 reveals stronger effects than the ViT.

| | Settings | Test Accuracy | | |
|---|---|---|---|---|
| $\varphi$ | Terms | Min | Median | Max |
| 1.0 | $\rho$ | 90.6% | 91.2% | 91.5% |
| 1.0 | $h_{\mathrm{abs}}$ | 89.7% | 90.1% | 90.8% |
| 1.0 | $\rho + h_{\mathrm{abs}}$ | 91.0% | 91.8% | 92.7% |
| 1.0 | $h_{\mathrm{rms}}$ | 90.5% | 90.8% | 91.2% |
| 0.1 | $\rho$ | 92.1% | **92.6%** | 92.7% |
| 0.1 | $h_{\mathrm{abs}}$ | 91.8% | 91.9% | 92.1% |
| 0.1 | $\rho + h_{\mathrm{abs}}$ | 92.1% | 92.4% | **92.8%** |
| 0.1 | $h_{\mathrm{rms}}$ | 91.8% | 92.3% | 92.7% |

| | Settings | Test Loss | | |
|---|---|---|---|---|
| $\varphi$ | Terms | Min | Median | Max |
| 1.0 | $\rho$ | 4.49 | 4.49 | 4.55 |
| 1.0 | $h_{\mathrm{abs}}$ | 4.45 | 4.49 | 4.73 |
| 1.0 | $\rho + h_{\mathrm{abs}}$ | 4.41 | 4.48 | 4.48 |
| 1.0 | $h_{\mathrm{rms}}$ | 4.46 | 4.47 | 4.65 |
| 0.1 | $\rho$ | 4.08 | 4.09 | 4.12 |
| 0.1 | $h_{\mathrm{abs}}$ | 4.13 | 4.13 | 4.14 |
| 0.1 | $\rho + h_{\mathrm{abs}}$ | **4.05** | **4.07** | 4.11 |
| 0.1 | $h_{\mathrm{rms}}$ | 4.14 | 4.15 | 4.15 |

Table 1: ResNet18 / CIFAR-10: comparison of curvature terms[2] under Nesterov acceleration.

Table 2: ViT / Imagenet: comparison of curvature terms[2] under Nesterov acceleration.

Taken together, these experiments validate the theoretical prediction that choosing $\varphi = 1 - \beta_1$ aligns the quasi-Newton update with the Nesterov evaluation point. The improvement appears specifically when using the algebraically matched NAQ fractions predicted by Theorem 6. We note that the relative strength of glass contributions differs systematically across architectures. In ResNet18, perturbations propagate through many stacked nonlinearities, yielding higher densities of activation-boundary crossings. In the ViT, glass effects are weaker, likely because residual connections limit the impact of any single-parameter perturbation: each block modifies only an additive component of the residual stream, and subsequent normalization further reduces its downstream influence.

**Exploration and Stability.** Theorem 5 predicts that curvature-aware steps can shrink optimally, but may also overshoot if not bounded. In practice, we observe that the raw quasi-Newton step can become excessively large and exceed the range where curvature extrapolation is reliable. To stabilize training, Alice applies step-length bounds $\lambda_{\min}$ and $\lambda_{\max}$, interpretable as Adam-style learning rates. Between these bounds, NAQ steps are used. Conveniently, by setting $\varphi = \omega = 1$ and $\lambda_{\min} = \lambda_{\max} =$ learn-rate recovers Adam exactly, so hyperparameters exist for which Alice will not underperform this baseline. Figure 5 illustrates this effect using the Tensorized Transformer (Ma et al., 2019) on WikiText-103 (Merity et al., 2016): increasing $\lambda_{\max}$ accelerates early loss reduction, confirming the value of stability constraints. Table 3 provides additional statistics. Results show that the stability bound $\lambda_{\max}$ is the decisive factor: all curvature terms achieve markedly lower perplexity when $\lambda_{\max}$ is increased from $1.0 \times 10^{-3}$ to $2.5 \times 10^{-3}$. Indeed, the minimum perplexity at the smaller bound is higher than the maximum perplexity at the larger bound in nearly every case. While different curvature computations shift the relative best values—the glass term $\rho$ attains the lowest absolute minimum and $h_{\mathrm{abs}}$ achieves the lowest median—the dominant effect comes from the exploration limit imposed by $\lambda_{\max}$. These findings highlight that step-size safeguards are not merely a precaution but a key enabler of stable and efficient curvature exploitation.

**Method Comparisons.** Finally, to situate curvature effects in familiar training scenarios, we compare Alice against standard baselines. Summary statistics over multiple seeds (min/median/max) are provided in Appendix C, showing consistent improvements within the usual variance of neural optimizers. Figure 6 reports results on ResNet18/CIFAR-10 and Tensorized Transformer/WikiText-103. On ResNet18 (a), Alice achieves substantially lower validation loss and higher validation accuracy than SGD with momentum, Adam, and AdaHessian, with the improvement over AdaHessian

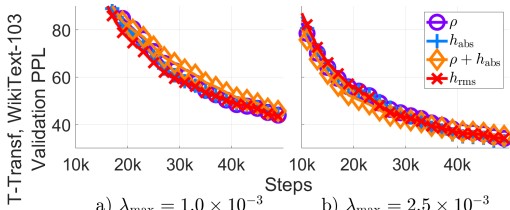

Figure 5: Effect of learning-rate bounds on stability and early loss reduction.

| Settings | | Validation PPL | | |
|---|---|---|---|---|
| $\lambda_{\max}$ | Terms | Min | Median | Max |
| $1.0 \times 10^{-3}$ | $\rho$ | 40.4 | 43.2 | 49.1 |
| $1.0 \times 10^{-3}$ | $h_{\mathrm{abs}}$ | 38.6 | 43.7 | 47.1 |
| $1.0 \times 10^{-3}$ | $\rho + h_{\mathrm{abs}}$ | 39.8 | 45.1 | 48.2 |
| $1.0 \times 10^{-3}$ | $h_{\mathrm{rms}}$ | 39.4 | 42.9 | 51.2 |
| $2.5 \times 10^{-3}$ | $\rho$ | **28.8** | 33.6 | 35.8 |
| $2.5 \times 10^{-3}$ | $h_{\mathrm{abs}}$ | 30.0 | **32.5** | 36.8 |
| $2.5 \times 10^{-3}$ | $\rho + h_{\mathrm{abs}}$ | 31.0 | 34.8 | 39.1 |
| $2.5 \times 10^{-3}$ | $h_{\mathrm{rms}}$ | 28.9 | 33.7 | 39.9 |

Table 3: Tensorized Transformer / WikiText-103: diagnostic comparison of curvature terms[2] under step-length bounds.

being particularly pronounced. On the Tensorized Transformer (b), we compare Adam, AdaHessian, and Alice with glass-only and Hessian-only curvature. Here, Alice with Hessian-based curvature slightly improves over Adam, and both outperform AdaHessian.

The purpose is not to claim optimization superiority, but to demonstrate that curvature-of-expectation can be extracted consistently across architectures and may yield measurable training benefits. This diagnostic role makes the influence of glass curvature empirically visible, and establishes a baseline for future work in pruning, quantization, and curvature-aware regularization.

## 5 Broader Implications of Curvature-of-Expectation

Curvature-of-expectation extends curvature methods beyond the Hessian, with glass curvature arising from cumulative gradient discontinuities in nonsmooth regions. Together these form a unified picture: Hessian and glass curvature are not competing notions, but limiting cases of one framework. This perspective clarifies why Hessian-based methods succeed in some settings but fail in others, and explains the variability long observed in their empirical performance.

**Compression and Pruning.** Diagonal Hessian approximations are widely used to estimate the effect of weight removal. Our results suggest that glass curvature contributes an additional diagonal term that can dominate when many activation boundaries intersect a parameter's influence. This explains mismatches between pruning theory and practice, and motivates new criteria that explicitly incorporate glass terms. Quantization also introduces discontinuities, thus curvature-of-expectation may inform when quantization schemes destabilize training and how stability can be recovered. Leveraging glass density for these purposes is a natural next step, but requires a dedicated study beyond the present scope.

**Generalization and Flatness.** Hessian eigenvalues are often proxies for sharpness, yet their relation to generalization remains debated. Expectation curvature offers an alternative: glass terms bound expected loss increases with a $3/2$ power law, distinct from quadratic Hessian growth. This provides a sharper flatness diagnostic for non-smooth regions and could serve as a complementary metric.

**Interpretation with Alice.** Because Alice isolates curvature contributions, it enables practitioners to test directly when glass curvature is negligible, comparable, or dominant. If glass terms vanish, Hessian-based methods suffice. If they dominate, one may prefer smoother activations or rely on the stabilizing influence of the modified quasi-Newton update. Either way, Alice provides a diagnostic baseline that translates theoretical distinctions into actionable practice.

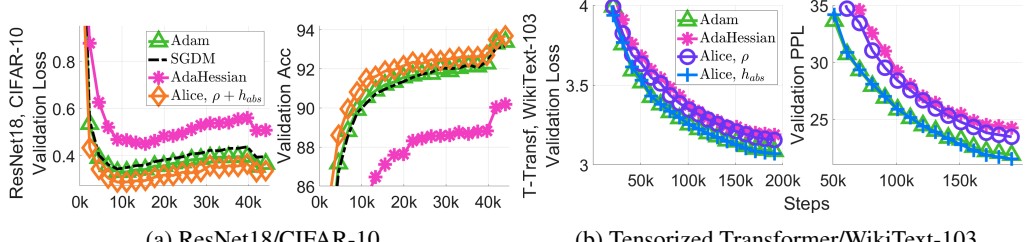

(a) ResNet18/CIFAR-10      (b) Tensorized Transformer/WikiText-103

Figure 6: Comparisons with standard optimizers provide context for curvature terms[2]. Alice is used diagnostically to test the role of Hessian vs. glass curvature.

| Method | Test Accuracy | | |
|---|---|---|---|
| | Min | Median | Max |
| Adam | 93.23% | 93.41% | 93.73% |
| SGDM | 92.65% | 93.19% | 93.75% |
| AdaHessian | 78.41% | 90.25% | 91.50% |
| Alice, $\rho + h_{\mathrm{abs}}$ | 93.30% | **93.69%** | **94.09%** |

Table 4: ResNet18 / CIFAR-10: diagnostic comparison with standard optimizers.

| Method | Validation PPL | | |
|---|---|---|---|
| | Min | Median | Max |
| Adam | 20.84 | 21.83 | 21.86 |
| AdaHessian | 22.12 | 24.23 | 25.79 |
| Alice, $\rho$ | 21.07 | 23.43 | 24.95 |
| Alice, $h_{\mathrm{abs}}$ | **18.66** | **21.50** | 23.74 |

Table 5: Tensorized Transformer / WikiText-103: curvature diagnostic comparisons.

## 6 CONCLUSION

We developed a theoretical framework for curvature-of-expectation in nonsmooth loss landscapes, where activation-induced gradient discontinuities accumulate into a dense field, i.e. a *gradient glass*. Our analysis introduced: (i) a model of gradient glass and its density, (ii) optimal kernels and perturbation distributions for diagonal estimation, (iii) bounds on expected loss change with a $3/2$ power law, (iv) quasi-Newton steps that unify Hessian and glass curvature, and (v) exactness conditions for Nesterov acceleration. To translate these contributions into practice, we built ALICE, a streamlined optimizer designed as an investigative probe. Alice isolates Hessian and glass curvature effects by inserting their estimates into controlled updates, thereby showing when and how curvature extraction alters training dynamics. Our experiments confirm that curvature-of-expectation is measurable, relevant, and capable of influencing stability and effective step size. In this way, Alice is not a general-purpose optimizer but a scientific instrument: it validates that curvature-of-expectation is measurable, relevant, and capable of guiding training. By making these effects visible, Alice lays a reproducible foundation for a new research direction in optimization that builds on a consistent theoretical picture to enable curvature-aware methods across pruning, quantization, and regularization.

## 7 LIMITATIONS

Our framework and probe introduce several constraints. First, extracting curvature requires three gradient evaluations per batch. This adds overhead, though far less than full second-order back-propagation, and curvature terms evolve slowly enough to amortize cost across steps. Second, our analysis assumes gradient discontinuities from ReLU-like units are approximately uniform at the chosen perturbation scale. This captures large networks well, but smoother activations may diminish the glass effect. Third, Alice includes hyperparameters for perturbation radius and step-size bounds. These are interpretable extensions of Adam defaults and need only modest tuning, but they add knobs that must be set. These constraints reflect design tradeoffs, not conceptual limits. Alice demonstrates that glass curvature is empirically accessible, leaving open how future optimizers may exploit it efficiently. Together, these results show that curvature-of-expectation offers a principled and practical complement to the Hessian, revealing structure in loss landscapes that was previously hidden.

**Use of LLMs.** We used ChatGPT (GPT-5, OpenAI) to assist with improving the clarity and consistency of exposition, formatting LaTeX, and checking grammar. All ideas, experiments, and analyses were conceived, designed, and executed by the authors.

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

The authors used an AI-assisted writing tool (OpenAI's ChatGPT) for language editing and clarity improvements. All conceptual contributions, experiments, and programming were carried out by the authors.

## A  PROOFS

**Proof of Theorem 1 (Glass from ReLUs).**  For a unit $k \in \mathcal{S}_\Delta$, write $\Delta_k = \boldsymbol{\delta}^\top \hat{\boldsymbol{\gamma}}^{(k)}$. With uniform density on $[-\Delta, \Delta]$, the probability of a flip is

$$p(\text{flip } k) = \frac{|\Delta_k|}{2\Delta}. \tag{17}$$

Given $z^{(k)} = \max(0, y^{(k)})$, a flip changes $\frac{\partial z^{(k)}}{\partial y^{(k)}}$ by $\pm 1$. By the chain rule, the gradient changes by $\gamma^{(k)} = \pm \frac{d\mathcal{L}}{dz^{(k)}} \hat{\boldsymbol{\gamma}}^{(k)}$. By the independence assumption,

$$\boldsymbol{v}_i(\boldsymbol{\delta}) = \sum_{k \in \mathcal{S}_\Delta} p(\text{flip } k) \left( \frac{d\mathcal{L}}{dz^{(k)}} \right)^2 \hat{\boldsymbol{\gamma}}_i^{(k)2}.$$

Finally, $|\Delta_k| \le \sum_j |\hat{\boldsymbol{\gamma}}_j^{(k)}| \, |\boldsymbol{\delta}_j|$ gives the stated bound in Equation (3). ∎

**Proof of Theorem 2.**  Write $\boldsymbol{y}_i = \sum_j \boldsymbol{M}_{ij} \boldsymbol{\delta}_j$. Since $\mathbb{E}[\boldsymbol{\delta}_j] = 0$, $\mathbb{E}[\boldsymbol{\delta}_j^2] = 1$, and $\mathbb{E}[\boldsymbol{\delta}_i \boldsymbol{\delta}_j] = 0$ for $i \ne j$, integrating over all $\{\boldsymbol{\delta}_j\}_{j \ne i}$ gives

$$\int \{\boldsymbol{y}_i, \, \boldsymbol{y}_i^2\} \prod_{j \ne i} dp(\boldsymbol{\delta}_j) = \{\boldsymbol{m}_i \, \boldsymbol{\delta}_i, \, \boldsymbol{m}_i^2 (\boldsymbol{\delta}_i^2 + \boldsymbol{\omega}_i^2)\}.$$

Consider estimators $Z_i = \kappa_i(\boldsymbol{\delta}_i) \, \boldsymbol{y}_i$. The zero-bias condition Equation (4) is equivalent to

$$\int \kappa_i(\boldsymbol{\delta}_i) \, \boldsymbol{\delta}_i \, dp(\boldsymbol{\delta}_i) = 1. \tag{18}$$

The variance is then

$$\text{Var}[Z_i] = \mathbb{E}[Z_i^2] - \boldsymbol{m}_i^2 = \boldsymbol{m}_i^2 \left( \int \kappa_i(\boldsymbol{\delta}_i)^2 \, (\boldsymbol{\delta}_i^2 + \boldsymbol{\omega}_i^2) \, dp(\boldsymbol{\delta}_i) - 1 \right).$$

To minimize this subject to Equation (18), apply the variational principle. Let $\kappa_i = \kappa_i^* + \varepsilon \, \eta$ where $\eta$ satisfies $\int \eta(\boldsymbol{\delta}_i) \, \boldsymbol{\delta}_i \, dp(\boldsymbol{\delta}_i) = 0$. Differentiating at $\varepsilon = 0$ yields

$$\left. \frac{\partial \text{Var}[Z_i]}{\partial \varepsilon} \right|_{\varepsilon = 0} = 2\boldsymbol{m}_i^2 \int \eta(\boldsymbol{\delta}_i) \, \kappa_i^*(\boldsymbol{\delta}_i) \, (\boldsymbol{\delta}_i^2 + \boldsymbol{\omega}_i^2) \, dp(\boldsymbol{\delta}_i) = 0.$$

Thus $\kappa_i^*(\boldsymbol{\delta}_i) \, (\boldsymbol{\delta}_i^2 + \boldsymbol{\omega}_i^2) \propto \boldsymbol{\delta}_i$, i.e. $\kappa_i^*(\boldsymbol{\delta}_i) \propto \boldsymbol{\delta}_i / (\boldsymbol{\delta}_i^2 + \boldsymbol{\omega}_i^2)$. The normalization is fixed by Equation (18). Uniqueness holds in the weighted $L^2$ class since the objective is strictly convex under weight $(\boldsymbol{\delta}_i^2 + \boldsymbol{\omega}_i^2)$ and the constraint space is affine. ∎

**Proof of Theorem 3.**  Let $p^*$ denote the optimal factor distribution and write a perturbation $p(\boldsymbol{\delta}_i) = p^*(\boldsymbol{\delta}_i) + \varepsilon \, \eta(\boldsymbol{\delta}_i)$, restricted so that $\eta(\boldsymbol{\delta}_i) = 0$ when $p^*(\boldsymbol{\delta}_i) = 0$. Normalization requires

$$\int \eta(\boldsymbol{\delta}_i) \, d\boldsymbol{\delta}_i = 0. \tag{19}$$

With the optimal kernel Equation (5), the variance takes the form

$$\boldsymbol{m}_i^{-2} \text{Var}[Z_i] = \int \kappa_i^*(\boldsymbol{\delta}_i)^2 \, (\boldsymbol{\delta}_i^2 + \boldsymbol{\omega}_i^2) \, dp(\boldsymbol{\delta}_i) - 1 = c^{-1} - 1,$$

where $c = \int \frac{\boldsymbol{\delta}_i^2}{\boldsymbol{\delta}_i^2 + \boldsymbol{\omega}_i^2} \, dp(\boldsymbol{\delta}_i)$. Differentiating w.r.t. $\varepsilon$ gives

$$\left. \frac{\partial}{\partial \varepsilon} \text{Var}[Z_i] \right|_{\varepsilon = 0} = -\boldsymbol{m}_i^2 c^{-2} \int \frac{\boldsymbol{\delta}_i^2}{\boldsymbol{\delta}_i^2 + \boldsymbol{\omega}_i^2} \, \eta(\boldsymbol{\delta}_i) \, d\boldsymbol{\delta}_i = 0.$$

Thus, on the support of $p^*$ the function $\frac{\delta_i^2}{\delta_i^2 + \omega_i^2}$ must be constant, since Equation (19) provides the only restriction on $\eta$ there. This is possible only if the support consists of two symmetric points $\{\pm x\}$. Imposing zero mean forces symmetry, and unit variance requires $x = 1$, giving the Rademacher distribution. $\blacksquare$

**Proof of Theorem 4.** Fix a coordinate $k$. Divide the path $\boldsymbol{\mu}_k \to \boldsymbol{\mu}_k + \boldsymbol{\delta}_k$ into $n$ segments. Let $\boldsymbol{\gamma}_k^{(j)}$ be i.i.d. zero-mean increments with $\mathbb{E}[(\boldsymbol{\gamma}_k^{(j)})^2] = \boldsymbol{\rho}_k|\boldsymbol{\delta}_k|/n$ so that density $\boldsymbol{\rho}_k$ is preserved. The loss change before applying the floor is

$$\Delta_n(\boldsymbol{\delta}_k) = \sum_{j=1}^{n} \frac{n-j}{n} \boldsymbol{\gamma}_k^{(j)} \boldsymbol{\delta}_k.$$

Each term has mean zero, so variances add:

$$\mathbb{E}[\Delta_n(\boldsymbol{\delta}_k)^2] = \sum_{j=1}^{n} \left(\frac{n-j}{n}\right)^2 \frac{\boldsymbol{\rho}_k|\boldsymbol{\delta}_k|}{n} \boldsymbol{\delta}_k^2.$$

Taking $n \to \infty$ gives

$$\mathbb{E}[\Delta(\boldsymbol{\delta}_k)^2] = \tfrac{1}{3} \boldsymbol{\rho}_k|\boldsymbol{\delta}_k|^3.$$

By the Lyapunov CLT, $\Delta(\boldsymbol{\delta}_k) \sim \mathcal{N}\left(0, \boldsymbol{\rho}_k|\boldsymbol{\delta}_k|^3/3\right)$. Imposing the floor $\Delta\mathcal{L}(\boldsymbol{\delta}_k) = |\Delta(\boldsymbol{\delta}_k)|$ makes this a folded normal, whose mean is $\sqrt{2/\pi}\,\sigma$ with variance parameter $\sigma^2$. Summing over coordinates yields the bound in Equation (6). $\blacksquare$

**Proof of Theorem 5.** The bound separates by coordinate $k$. Stationarity requires

$$0 = \boldsymbol{g}_k + \boldsymbol{h}_k \boldsymbol{\delta}_k + \sqrt{\tfrac{3\boldsymbol{\rho}_k}{2\pi}} \ \text{sign}(\boldsymbol{\delta}_k)|\boldsymbol{\delta}_k|^{1/2}.$$

To enforce descent, set $\boldsymbol{\delta}_k = -\text{sign}(\boldsymbol{g}_k)x^2$ with $x > 0$. Substituting $\hat{\boldsymbol{h}}_k = 3\boldsymbol{\rho}_k/(4\pi|\boldsymbol{g}_k| + \varepsilon)$ gives

$$0 = -\sqrt{\tfrac{|\boldsymbol{g}_k|}{2}} + \sqrt{\hat{\boldsymbol{h}}_k}\,x + \frac{\boldsymbol{h}_k}{\sqrt{2|\boldsymbol{g}_k|}}\,x^2.$$

Solving the quadratic (positive root) yields

$$x = \frac{\sqrt{2|\boldsymbol{g}_k|}}{\sqrt{\hat{\boldsymbol{h}}_k} + \sqrt{\hat{\boldsymbol{h}}_k + 2\boldsymbol{h}_k}}.$$

Thus $\boldsymbol{\delta}_k = -\boldsymbol{g}_k/(\hat{\boldsymbol{h}}_k + \boldsymbol{h}_k + \sqrt{\hat{\boldsymbol{h}}_k(\hat{\boldsymbol{h}}_k + 2\boldsymbol{h}_k)} + \varepsilon)$, which matches Equation (8). $\blacksquare$

**Proof of Theorem 6.** From Equation (9) and Equation (11),

$$g(\boldsymbol{\mu}^{(s+1)}) = \boldsymbol{g}^{(s)} + \varphi\bar{\boldsymbol{h}} \odot \boldsymbol{\delta}^{(s)} = \boldsymbol{g}^{(s)} - (\varphi)\boldsymbol{g}^{(s)} = (1 - \varphi)\boldsymbol{g}^{(s)}.$$

Thus setting $\varphi = 1 - \beta_1$ makes $g(\boldsymbol{\mu}^{(s+1)}) = \beta_1 \boldsymbol{g}^{(s)}$, consistent with Equation (10). Next, substitute the evaluation point $\omega = 1$ into Equation (12), combine with the error decomposition Equation (13), and place the result in Equation (10):

$$\boldsymbol{g}^{(s+1)} = \beta_1 \boldsymbol{g}^{(s)} + (1 - \beta_1)\big(\boldsymbol{g}^{(s)} - \boldsymbol{\gamma}^{(s)} + \boldsymbol{H}\boldsymbol{\delta}^{(s)}\big)$$
$$= \boldsymbol{g}^*(\boldsymbol{\mu}^{(s+1)}) + \beta_1 \boldsymbol{\gamma}^{(s)}.$$

This shows that the update both tracks the hidden linear structure and contracts errors by $\beta_1$. $\blacksquare$

## B    METHODOLOGICAL DETAILS

**Glass Illustration (Fig. 1).** We sample $n$ random boundary anchors $b_k \in [0,1]^2$, apply a light repulsive update to spread them uniformly, and assign each a random normalized gradient jump $g_k$ and a random activation phase sign $s_k \in \{\pm 1\}$. Given two trajectory points $t_0, t_1$, we discretize the line segment and track each boundary crossing where $\text{sign}((t - b_k)^\top g_k)$ changes. The gradient along the trajectory accumulates jumps $s_k g_k$ at each crossing, and the loss is obtained by integrating these gradient values. The glass density $\rho$ is computed as the cumulative second moment of gradient jumps divided by the trajectory length, which yields the envelopes in the right panel.

---

**Algorithm 2** Alice Diagnostic Topography Update

**Input**: evaluation center $\boldsymbol{\nu}$; gradient function $g(\boldsymbol{\theta})$.
**Input/Output**: running averages $\boldsymbol{g}$, $\boldsymbol{\rho}$, $\boldsymbol{h}_{\mathrm{abs}}$, $\boldsymbol{h}_{\mathrm{rms}}^2$, and $\boldsymbol{s}$.
**Hyperparameters**: $\lambda, \beta_1, \beta_2$.
Implements Eqs. 14–16 to adapt Theorems 1–3, requiring only one parameter-length temporary variable $\boldsymbol{t}$.

1: Draw Rademacher $\boldsymbol{t}$. Set $\boldsymbol{\theta} \leftarrow \boldsymbol{\nu} + \lambda \boldsymbol{t}$.
2: Evaluate $g(\boldsymbol{\theta}) = \boldsymbol{g}^{(+)}$.
3: Set $\boldsymbol{\theta} \leftarrow \boldsymbol{\nu} - \lambda \boldsymbol{t}$. Store $\boldsymbol{t} \leftarrow \boldsymbol{g}^{(+)}$.
4: Evaluate $g(\boldsymbol{\theta}) = \boldsymbol{g}^{(-)}$.
5: $\boldsymbol{h}_{\mathrm{abs}} \leftarrow \beta_2 \boldsymbol{h}_{\mathrm{abs}} + (1 - \beta_2)\frac{1}{2\lambda}|\boldsymbol{t} - \boldsymbol{g}^{(-)}|$.
6: $\boldsymbol{h}_{\mathrm{rms}}^2 \leftarrow \beta_2 \boldsymbol{h}_{\mathrm{rms}}^2 + (1 - \beta_2)\frac{1}{4\lambda^2}(\boldsymbol{t} - \boldsymbol{g}^{(-)})^2$.
7: Store $\boldsymbol{t} \leftarrow \frac{1}{2}(\boldsymbol{t} + \boldsymbol{g}^{(-)})$. Set $\boldsymbol{\theta} \leftarrow \boldsymbol{\mu}$.
8: Evaluate $g(\boldsymbol{\theta}) = \boldsymbol{g}^{(0)}$.
9: $\boldsymbol{g} \leftarrow \beta_1 \boldsymbol{g} + (1 - \beta_1)\boldsymbol{g}^{(0)}$.
10: $\boldsymbol{\rho} \leftarrow \beta_2 \boldsymbol{\rho} + (1 - \beta_2)\frac{2}{\lambda}(\boldsymbol{t} - \boldsymbol{g}^{(0)})^2$.
11: $\boldsymbol{s} \leftarrow \beta_2 \boldsymbol{s} + (1 - \beta_2)\boldsymbol{g}^{(0)2}$.

---

---

**Algorithm 3** Alice Quasi-Newton Step with Diagnostic Curvature Terms

**Input**: running averages $\boldsymbol{g}$, $\boldsymbol{\rho}$, $\boldsymbol{h}$, $\boldsymbol{s}$.
**Input/Output**: parameters $\boldsymbol{\mu}$.
**Output**: evaluation center $\boldsymbol{\nu}$.
**Hyperparameters**: limit-method, $\lambda, \lambda_{\min}, \lambda_{\max}, \varepsilon$.
Implements Theorems 4, 5, and 6, with stability enforced by Adam-style limits.

1: Compute glass term $\hat{\boldsymbol{h}} = \frac{3}{4\pi}\boldsymbol{\rho} \odot (|\boldsymbol{g}| + \varepsilon)^{-1}$.
2: Compute modified Hessian $\bar{\boldsymbol{h}} = \hat{\boldsymbol{h}} + \boldsymbol{h} + \sqrt{\hat{\boldsymbol{h}} \odot (\hat{\boldsymbol{h}} + 2\boldsymbol{h})} + \varepsilon$.
3: Quasi-Newton scale $\boldsymbol{\delta} = |\boldsymbol{g}| \odot \bar{\boldsymbol{h}}^{-1}$.
4: Apply limits:  Fixed scale: $\boldsymbol{\delta}_{\min/\max} = \lambda_{\min/\max}$;  SGD-M: $\boldsymbol{\delta}_{\min/\max} = \lambda_{\min/\max}|\boldsymbol{g}|$;
   Adam: $\boldsymbol{\delta}_{\min/\max} = \lambda_{\min/\max}|\boldsymbol{g}| \odot (\sqrt{\boldsymbol{s}} + \varepsilon)^{-1}$.
5: Enforce bounds: $\boldsymbol{\delta} \leftarrow \max(\boldsymbol{\delta}_{\min}, \min(\boldsymbol{\delta}_{\max}, \boldsymbol{\delta}))$.
6: Correct sign: $\boldsymbol{\delta} \leftarrow -\operatorname{sign}(\boldsymbol{g})\boldsymbol{\delta}$.
7: Update evaluation center $\boldsymbol{\nu} \leftarrow \boldsymbol{\mu} + \omega\boldsymbol{\delta}$.
8: Update parameters $\boldsymbol{\mu} \leftarrow \boldsymbol{\mu} + \varphi\boldsymbol{\delta}$.

---

**Algorithms.** Algorithm 2 shows the full topography update in a memory-efficient form using one temporary variable. Line 6 additionally enables an RMS Hessian approximation: taking the square root of the running second moment yields a non-negative diagonal suitable for quasi-Newton steps. This is similar to AdaHessian (Yao et al., 2021) and Sophia (Liu et al., 2023), except that we maintain per-parameter statistics instead of block averages. Algorithm 3 then applies quasi-Newton updates combining glass and Hessian diagonals, with bounds for stability.

# C  EXPERIMENTAL SETUP

We evaluate Alice as a diagnostic probe on standard benchmarks. Results are reported as minimum, median, and maximum values over multiple seeds, showing both variability and best-case outcomes.

**Hardware and Software.** Experiments were run on compute nodes with AMD 32-core CPUs and one NVIDIA A100 GPU (40GB memory). Code is implemented in PyTorch 2.2.1 (BSD 3-Clause License) with CUDA 11.8.

**Licenses.** PyTorch: BSD 3-Clause. ResNet: BSD 3-Clause. Tensorized Transformer and CIFAR-10: MIT. WikiText-103 (and Wikipedia): CC BY-SA 3.0. ImageNet: non-commercial research/educational use only.

**ResNet18, Power Law.** Figure 2 uses a specialized Alice variant with additional memory and evaluations to store gradient variations at $\lambda$ and $2\lambda$. We train a modified ResNet18 with multiplicative channel masks, improving prediction quality for SGDM, Adam, and Alice. Training: 40 epochs, $\lambda = 0.002$, $\beta_1 = 0.9$, $\beta_2 = 0.999$, $\varepsilon = 10^{-8}$, $\varphi = 0.1$, $\omega = 1.0$, $\lambda_{\min} = 0$, $\lambda_{\max} = 0.002$, quick_steps $= 0$, Adam-based limiting. Curvature terms: $\boldsymbol{\rho}$ and $\boldsymbol{h}_{\mathrm{abs}}$.

**ResNet18, NAQ.** Figure 4: ResNet18 with tuned $\lambda = 0.005$ and fixed-scale limiting ($\lambda_{\max} = 5 \times 10^{-3}$). Grid search over $\{0.001, 0.002, 0.005, 0.01, 0.02, 0.05\}$ and limiting strategies. 10 seeds per setting.

**ViT, NAQ.** Vision Transformer with 8 layers, linear-complexity attention (Shen et al., 2021), Imagenet-64x64. Training: 4 epochs, $\lambda = 0.005$, $\beta_1 = 0.9$, $\beta_2 = 0.999$, $\varepsilon = 10^{-8}$, $\lambda_{\min} = 0$, $\lambda_{\max} = 0.01$, quick_steps $= 3$, Adam-based limiting. 5 seeds per setting.

**Tensorized Transformer, Stability Limit.** Tensorized Transformer (Ma et al., 2019), dropout 0.1, batch size 60, 50k steps. Settings: $\lambda = 0.001$, $\beta_1 = 0.9$, $\beta_2 = 0.999$, $\varepsilon = 10^{-8}$, $\varphi = 0.1$, $\omega = 1.0$, quick_steps $= 3$, Adam-based limiting.

**ResNet18, Training Methods.** For Figure 6: Adam ($\lambda = 0.001$, $\beta_1 = 0.9$, $\beta_2 = 0.999$, $\varepsilon = 10^{-8}$); SGD-M ($\lambda = 0.001$, $\beta = 0.9$); AdaHessian ($\lambda = 0.15$, tuned from $\{0.1, 0.15, 0.2, 0.25, 0.3\}$); Alice ($\lambda = 0.005$, $\beta_1 = 0.9$, $\beta_2 = 0.999$, $\varepsilon = 10^{-8}$, $\varphi = 0.1$, $\omega = 1.0$, $\lambda_{\min} = 0$, $\lambda_{\max} = 0.01$, quick_steps $= 3$, Adam-based limiting). Curvature: $\boldsymbol{\rho}$ and $\boldsymbol{h}_{\mathrm{abs}}$.

**Tensorized Transformer, Training Methods.** For Figure 6: 200k steps. Adam and Alice: $\lambda = 0.00025$, $\beta_1 = 0.9$, $\beta_2 = 0.999$, $\varepsilon = 10^{-8}$. Alice also: $\varphi = 1.0$, $\omega = 1.0$, $\lambda_{\max} = 0.000375$, quick_steps $= 1$, Adam-based limiting. AdaHessian: $\lambda = 0.00025$.

