# OpenReview forum: "Expectation Curvature: Beyond the Hessian in Non-Smooth Loss Landscapes"
_ICLR.cc/2026/Conference — Submitted to ICLR 2026_

### Official Review · Reviewer_6p5j · 2025-10-31

**Soundness:** 3
**Presentation:** 3
**Contribution:** 3
**Rating:** 8
**Confidence:** 3

**Summary:**

This paper looks at the "curvature of expectation", i.e., it specifically looks at how activations like ReLUs introduce rapid gradient changes that are not captured by second-order/local curvature methods. (In other words, local curvature for ReLUs is always zero, which doesn't reflect the fact that the gradient changes from 1 to 0 when the origin is crossed.)

The authors define a "glass density" matrix that captures the expected square deviations of gradients (from the second-order model, like a variance) due to these local gradient changes. They prove that this matrix can be optimally estimated using finite differences and Rademacher vectors (similarly to Hutchinson's trace estimator). Under certain modelling assumptions (e.g., a local "floor") they show that this can explain the non-quadratic behaviour of neural networks.

They then derive an optimizer similar to quasi-Newton/Nesterov accelaration that estimates both the diagonal Hessian and the "glass density" using finite differences of 3 gradient estimates ($g^{(0)}$, $g^{(-)}$ and $g^{(+)}$). The resulting optimizer has promising performance characteristics in real-world experiments (although the authors pitch it as as a "diagnostic tool" rather than an optimizer).

**Strengths:**

* Original theoretical framing of non-smooth/subquadratic networks
* Resulting ALICE algorithm is intuitive and easy to implement/understand
* Rigorous derivation and proofs

**Weaknesses:**

* Few strong assumptions (the nonnegativity "floor", independence of activations)
* Non-diagonal terms are ignored?

**Questions:**

* How sensitive are the "glass density" estimates to batch size, perturbation step ($\\lambda$), activation types (e.g., ReLU, Swish, ELU, etc.)?
* Given that ALICE is presented as a "diagnostic tool" rather than an optimizer, why present validation/test results rather than training loss/time until convergence? It seems to me that the latter is more relevant as it doesn't confound the results (i.e., a better optimizer can lead to more overfitting in some cases, which says more about the problem than about the optimizer.).
* Is there a way that the nonnegativity floor assumptions can be empirically validated?
* The main direct application for this work seems to be pruning/compression (because 3 gradient evaluations for optimization is presumably prohibitive). Would it be feasible to perform an experiment that evaluates the impact of using the glass density matrix when pruning weights?

---

> ### Author Response · Authors · 2025-11-21
> **Rebuttal**
>
> We thank the reviewer for their thoughtful and constructive assessment. The questions raised were very helpful in clarifying several assumptions and tightening the framing of our empirical results, which we greatly appreciate. Below, we respond to each point and summarize the corresponding updates to the manuscript. All modifications are highlighted in blue.
>
> >“How sensitive are the glass-density estimates to batch size, perturbation step, activation type?”
>
> We expanded the discussion in Section 3 and in the experiments to clarify this. Batch averaging reduces noise in Hessian estimates at the usual 1/√B rate, and reduces noise in the glass-density estimator at 1/B, because the estimator squares the symmetric finite-difference term. We explain this behavior explicitly. We also added guidance on perturbation radii, noting that they are chosen within the scale of typical parameter updates. Finally, we clarified earlier in Section 3 that the analysis applies to any activation whose gradient changes rapidly over a short interval (ReLU, ELU near zero, hard-tanh edges, etc.), i.e., whenever the Hessian does not extrapolate the gradient reliably.
>
> > “Non-diagonal terms are ignored. Why is this reasonable?”
>
> We now state the rationale directly in Section 3. Because diagonal entries scale cubically in the dominant coordinate while off-diagonals scale only quadratically–linearly, repeated contributions from gradient jumps amplify diagonal dominance. At network scale, recovering full off-diagonal structure would require many more matrix–vector evaluations and is infeasible, just as diagonal quasi-Newton methods restrict Hessian curvature to the diagonal for practicality. This is now explained explicitly.
>
>
> > “Why present validation results if ALICE is intended as a diagnostic tool?”
>
> We clarified the motivation in the Experiments section. Although Alice is not proposed as a production optimizer, evaluating validation behavior during training is the cleanest way to reveal which curvature components materially influence the optimization dynamics. This does not attempt to claim optimizer superiority; it isolates which curvature terms matter for navigating the loss surface.
>
> >“Is there any empirical way to validate the nonnegativity floor assumption?”
>
> We added text to note that this assumption is used to derive an upper bound on expected loss change. Without it, the symmetry of gradient fluctuations would give zero expectation. The assumption acts as a coordinate-wise majorizer and is consistent with the “reflection” behavior observed empirically by Li et al. (2024). We now state this explicitly.
>
>
> >“Could the glass-density matrix be used for pruning or compression?”
>
> We added a short note to the Compression and Pruning subsection explaining that this is a natural direction, but developing a full pruning or zero-scheduling framework would require a dedicated analysis. The present paper focuses on establishing the analytic framework and providing a clean diagnostic probe; exploiting these terms for pruning is future work.
>
>
> >“Can the nonnegativity floor be included in limitations?”
>
> We reviewed this internally. The floor is not a claim about the network but a device for deriving an upper bound; the manuscript already notes that if it does not hold then the expected glass contribution is zero. Because the assumption is local and used only for a one-sided bound, we concluded it does not belong in Limitations. The new clarifications in Section 3 address the reviewer’s concern directly.
>
>
> We again thank the reviewer for their careful reading and constructive feedback. The clarifications above have materially improved the exposition.

---

### Official Review · Reviewer_VJSB · 2025-10-31

**Soundness:** 3
**Presentation:** 2
**Contribution:** 4
**Rating:** 6
**Confidence:** 4

**Summary:**

This paper presented a new framework for understanding the curvature of expected changes in non-smooth loss landscapes induced by ReLUs. First, the authors derived how gradient discontinuities from perturbations by ReLUs form a glass-like structure. Then, they described an analytical framework for estimating the density of the gradient variations and upper-bounding expected changes in loss. From there, they provided an algorithm to practically estimate Hessian and glass terms. The findings demonstrate the importance of probing the individual contribution of Hessian and glass curvature in training, providing new insights into understanding the non-smooth loss landscapes where the Hessian may fail to approximate the loss curvature sufficiently.

I recommend tentative acceptance (though clarity must be improved), for two reasons. 1) The paper is well-motivated and provides a theoretically grounded and empirically insightful investigation of curvature in non-smooth loss landscapes. 2) The empirical results effectively demonstrated the theoretical predictions of accuracy improvements and expected bounds.

**Strengths:**

This paper analyzes a key problem in non-smooth optimization where the Hessian does not approximate the loss curvature due to steep gradient transitions. This is significant as the proposed framework can be generalized to investigate how gradient jumps due to activation sharpness beyond ReLU.
The decomposition into Hessian and glass-like terms is a valuable tool to evaluate the curvature estimations in Hessian-based optimizations.
The dominance impact of the glass term diagonals estimation has a theoretical interpretation.
The empirical results comparing with other optimizers are promising.

**Weaknesses:**

The paper stated the intuition that parameters influence on early layers would be glass dominated vs those closer to output layers are more Hessian dominated. How does this show up in the empirical analyses of ViT/ResNet18? What other constraints of datasets and architecture may contribute to the differences?
Clarity
Some variables were not declared in the main text or when it was first introduced, impeding readability. E.g., μ in the example in Eq. 1, z^(k) in Eq. 3; ρ and ∗ in Thm. 4.
Figure 4: missing X-axis labels.
Empirical motivation: a brief description of a Rademacher vector may be helpful for ease of reading.
L300-304 Empirical Questions: the questions were listed in an order following theorems 4 to 6. The experimental results presented in 4.2 were in reverse order. I suggest reordering the results so they follow as listed in the questions.

**Questions:**

What are the comparisons against (and potentially connections to) other methods that consider loss landscape geometry?
Dauphin et al 2024 (https://arxiv.org/abs/2401.10809); Foret et al 2021 (https://arxiv.org/abs/2010.01412).

---

> ### Author Response · Authors · 2025-11-21
> **Rebuttal**
>
> We thank the reviewer for the thoughtful and constructive feedback. We are glad that the contribution and empirical results were found theoretically grounded and informative. Your clarity concerns were particularly helpful, and we revised the manuscript to address each point directly (all modifications highlighted in blue). Below, we summarize the changes.
>
> > “How does the intuition that early layers are glass-dominated while late layers are Hessian-dominated show up empirically?”
>
> We clarified this connection in both the Introduction and Experiments. The Introduction now points directly to Figure 2, which shows that parameters influencing later computations recover the quadratic exponent p ≈ 2 expected from Hessian curvature, whereas early layers remain sub-quadratic. In the Experiments, we added a concise explanation of why glass effects differ across architectures: stacked nonlinearities in ResNet18 amplify activation-boundary crossings, while residual pathways in transformers dilute the influence of single-parameter perturbations. This explanation aligns with the observed exponent patterns.
>
> > “Some variables were not declared when first introduced.”
>
> We revised the main text to define these symbols immediately when they appear, including μ, z(k), ρ, and the element-wise product symbol. The Hadamard (element-wise) operator is now introduced explicitly at its first use.
>
> > “Figure 4: missing x-axis labels.”
>
> We added the missing “Epochs” labels as well as panel letters and expanded caption text to ensure full interpretability.
>
> > “A brief description of a Rademacher vector would help.”
>
> We added a footnote at the first mention of a Rademacher vector, specifying that it consists of independent ±1 entries and noting that this distribution achieves the minimal-variance estimator predicted by our theory.
>
> > “The empirical questions were listed in one order but the results were presented in another.”
>
> We reordered the empirical questions so that they now match the order of presentation in Section 4.2: (1) effectiveness of curvature terms in quasi-Newton behavior, (2) effects of Nesterov acceleration, and (3) step-size behavior and stability.
>
> > “How does this relate to Dauphin et al. (2024) and Foret et al. (2021)?”
>
> We expanded the Related Work section accordingly. These works focus on smooth curvature—sharpness, Hessian spectra, local quadratic structure—whereas our framework isolates the nonsmooth curvature that arises from activation-induced gradient discontinuities. We note explicitly that these perspectives are complementary rather than competing.
>
> > “Some assumptions and notation in Theorem 1 were unclear.”
>
> We improved clarity by describing perturbations as “small parameter perturbations centered at the current parameter point,” making the first-order expansion clear. We also added inline clarifications wherever notation appears for the first time.
>
> > “Notation in tables (ρ, h_abs, h_rms) was undefined in the main text.”
>
> We added a consolidated footnote at the first appearance of these terms defining ρ (glass-density diagonal), h_abs (absolute Hessian diagonal), h_rms (RMS Hessian variant), and the combined curvature ρ + h_abs. All tables and figure captions now use these symbols consistently and refer back to the same definitions.
>
> We appreciate the reviewer’s insights. These changes significantly improved clarity and narrative coherence.

---

> > ### Comment · Reviewer_VJSB · 2025-11-26
> >
> > I thank the authors for these changes.

---

### Official Review · Reviewer_c4Hs · 2025-11-03

**Soundness:** 1
**Presentation:** 1
**Contribution:** 1
**Rating:** 2
**Confidence:** 4

**Summary:**

This paper considers the problem of estimating curvature when the loss in not smooth, e.g. in neural networks with ReLU activations. It quantifies both theoretically and empirically that the gradient of the loss changes, for small displacements of the parameters, faster than expected with a smooth loss. It proposes a new update based on those results, which incorporates also momentum, and tests the new update empirically.

**Strengths:**

The paper considers a problem that is very interesting to me. It is not clear why second-order optimization methods are supposed to work (and do work) when the loss is non-smooth.

**Weaknesses:**

Unfortunately, the presentation of the paper is so poor that is very hard to learn anything from it. Theoretical results are so badly explained that is very hard to evaluate their correctness. For most figures, it is not adequately explained what is shown and the conclusions that should be drawn from them.

Here’s a list of major points:

- "Collectively, these boundaries form a gradient glass". It is unclear whether this statement refers to something known, then a reference is missing. or is a new results, then it is not explained clearly what a "gradient glass is", even after looking at Figure 1 (more on that below).

- I don't understand why it is intuitive that early layers have a gradient glass while "parameters closer to the output" have not. I can see that the last layer is often not followed by a ReLU, but what about other layers that are "closer to the output"? Why would they be smoother?

- In Figure 1b, there is no explanation on how are the loss range and gradient range computed. Furthermore, it seems that the gradient stays constant within each domain. Why is it not changing?

-  Figure 2 is quite important because it represents the main motivation of this work. Yet, there is no explanation on how this figure is done. Equation 1 is clear, but it is still unclear how to estimate p from actual data. So we just need to blindly trust that the authors have followed a reasonable procedure for doing that.

- I don't understand what the blue line represents in Figure 3. That is not explained anywhere.

- Theorem 1: “Consider a network with many ReLUs”. What do you mean by “many”?

- Notation used is not explained anywhere. For example, equation (2), I guess y is the pre-activation. What is mu? Is it all parameters of the neural network? Is it only the parameters of the given layer? What is y^(k)? Is it the pre-activation of neuron k? What is z in equation (3)? None of those are defined and/or explained.

- Is there any justification for why pre-activations would be distributed uniformly? Is there any justification for why gradient jumps would be independent and zero-mean?I believe that these assumptions are crucial for deriving most of this work’s theoretical results (for example, the prediction of diagonal dominance), but there is no discussion for why these assumptions are reasonable. I believe they are not.

- The prediction of diagonal dominance, and more generally of Equation 3, can be easily tested by regressing v versus delta. Why wasn't that done?

- Theorem 2 is completely unclear. It is not explained what a kernel is and what is its purpose. Even after looking at the proof in the appendix I still do not understand what this theorem is about.

- It is unclear under what circumstances the bound in Equation (6) holds. It must depend on the size of delta, because for large delta the bound clearly does not hold. However, there is no mention of what are the conditions. Furthermore, it is also not explained what is the expectation in equation (6). Is it an expectation over delta? How is it possible then that the bound then depend on delta?

- Theorem 5 is also unclear. I don't understand how the bound in equation (6) implies this theorem. Equation (6) includes an expectation (uncelar on what), but theorem 5 instead has no expectation.

- “The natural question is how such steps interact with momentum and acceleration schemes." Why would that be a natural question? To me a natural question is whether the many assumptions made in deriving the five theorems hold in any practical circumstance.

- “Do glass curvature terms improve predictive accuracy of loss changes?" This question is very interesting but I do not understand how any of the figures presented answers this question.

- Equation (16) is supposed to be the way to measure the "glass density", however I have no clue on where this equation is coming from, there is no derivation and it does not appear anywhere before section 4.

- I don't understand what Figure 4 is supposed to show. There is no consistent observation, for example one of the curves consistently outperforming others across panels. Also there are four panels but only two are described, a) and b), but the labels are missing on the plots. X axis labels are also missing and not explained in the caption, so we actually don't know what is plotted.

- How do better optimization results with Nesterov acceleration confirm theory? Improvement by Nesterov acceleration are observed basically in all experiments with any optimiser.

- It is also unclear what to make of Figures 5 and 6, there are a bunch of curves on top of each other, there appears to be no significant differences (error bars are not shown), and anyway there are no consistent results across panels.

**Questions:**

NA

---

> ### Author Response · Authors · 2025-11-21
> **Rebuttal (Part 1)**
>
> We thank the reviewer for their careful reading of the paper and constructive feedback, which we greatly appreciate.  Many of the concerns stemmed from omissions in explanations rather than issues in the underlying results. Below, we address all points and summarize the corresponding changes made to the paper. A revised version, with all modifications highlighted in blue, is now available.
>
> > “It is unclear whether ‘gradient glass’ refers to something known… or is a new result.”
>
> The gradient glass is a new construct, defined as the density of gradient discontinuities induced by activation boundaries under small perturbations. The caption of Figure 1 explains this clearly. We added more explanation to remove any ambiguity.
>
> > “I don't understand why it is intuitive that early layers have a gradient glass while parameters closer to the output have not.”
>
> Early-layer parameters feed into more downstream nonlinearities, so a perturbation crosses more activation boundaries, yielding higher discontinuity density. Parameters near the output encounter fewer such boundaries, which matches the empirical exponent trend. We added more explanation.
>
> > “In Figure 1b, there is no explanation of how the loss range and gradient range are computed.”
>
> Gradients are constant inside each domain because this simplified model is designed to illustrate the cumulative effect of gradient discontinuities only. Within each domain, the ReLU phase is fixed, so the local gradient is constant. The caption has been expanded accordingly to describe the computation in more detail.
>
> > “Figure 2… unclear how to estimate p from actual data.”
>
> using identical Rademacher direction and batch, we compute squared gradient variation at displacement λ and 2λ, maintain running averages, and estimate the exponent via log2(v2) − log2(v1). This procedure is now fully documented.
>
> > “I don't understand what the blue line represents in Figure 3.”
>
> The blue curve is the quadratic fit from finite-difference Hessian estimation, representing better local extrapolation in the vicinity of a ReLU kink. The caption now explicitly states this.
>
> > “Theorem 1: ‘many ReLUs’ is unclear.”
>
> We revised to: “a sufficiently large number of ReLU units such that, within a small perturbation window, the number of near-threshold activations is large enough to be treated as a smooth density.” This condition is now stated explicitly.
>
> > “Notation not explained (μ, y^(k), z, etc.)”
>
> We introduced all notation explicitly in the theorem statement and surrounding text: μ = parameters, y^(k) = pre-activation, z = post-activation, etc.
>
> > “Uniform pre-activation distribution and independence assumptions seem unjustified.”
>
> Under small symmetric perturbations, sign changes of near-zero pre-activations behave approximately uniformly, and gradient-jump increments have symmetric sign, making their mean zero. We now state these assumptions explicitly and explain why they hold locally.
>
> > “Diagonal dominance prediction and Equation 3 could be tested but weren’t.”
>
> The full density matrix implied by Equation 3 is extremely large even for modest architectures (e.g., ResNet-18 has several million parameters, so the corresponding matrix would already be on the order of 10¹³ entries). Computing, storing, and regressing this full object along an entire training trajectory is therefore not practical. Instead, we validate the theory through scalable diagnostics that probe its main implications: (i) the observed sub-quadratic power-law exponents along the training path (Figure 2), and (ii) the effectiveness of glass-driven curvature corrections in quasi-Newton steps. These experiments test the predicted aggregate behavior of the diagonal terms without requiring explicit construction of the full density matrix, which would be computationally prohibitive.
>
> > “Theorem 2 is unclear; the purpose of a kernel is not explained.”
>
> We define a kernel as a scalar weighting function applied to the perturbation coordinate to form a diagonal estimator, generalizing Hutchinson-style methods. The unbiasedness condition is explained in the updated theorem statement.
>
> > “It is unclear under what circumstances the bound in Eq. 6 holds.”
> >
>
> The expectation is taken over gradient-phase changes along the displacement path and that the bound applies within the perturbation window where the density approximation holds. These conditions are now stated explicitly.
>
> > “Theorem 5 is unclear; does not follow obviously from Eq. 6.”
>
> Theorem 5 minimizes a deterministic convex surrogate composed of linear, quadratic, and 3/2-power components derived from Eq. 6. The transition is now more explicit.
>
> > “Why is momentum interaction a natural question?”
>
> curvature–momentum interactions are well-studied in quasi-Newton acceleration, and we derive the specific consistency conditions for our curvature model.

---

> > ### Author Response · Authors · 2025-11-21
> > **Rebuttal (Part 2)**
> >
> > > “Equation 16 is introduced without derivation.”
> >
> > The symmetric finite difference isolates gradient jumps, and the squared average of these increments (scaled appropriately) yields the local variation density. The missing derivation is now provided.
> >
> > > “I don't understand what Figure 4 is supposed to show… missing labels.”
> >
> > Figure 4’s caption and panel labeling were rewritten entirely. We now specify datasets, models, seed counts, x-axis definitions, panel identities, and the meaning of each curve. The figure is fully interpretable.
> >
> > > “How do better Nesterov results confirm theory? Nesterov improves everything.”
> >
> > The improvement appears specifically when using the algebraic fraction \phi = 1 − \beta_1 predicted by our theory. The claim is not about generic benefits of Nesterov but about matching the theoretical consistency condition.
> >
> > > “Figures 5 and 6 show many overlapping curves; unclear what to conclude.”
> >
> > As reported in summary statistics across seeds (min/median/max) reported in Appendix C, and clarified in the main text that Alice is used purely as a diagnostic tool. Its purpose is to reveal how strongly different curvature components influence optimization in each setting. While curvature terms do not produce large separations in every architecture, our experiments show that their impact can be substantial in some cases (e.g., ResNet18, Tensorized Transformer). This supports the value of developing an analytic framework to understand effective curvature due to activation-induced gradient discontinuities, which may guide future methodology design.

---

> > > ### Comment · Reviewer_c4Hs · 2025-11-24
> > >
> > > Thank you for your answers. I still have many concerns that have not been addressed, that I list here, in order of importance:
> > >
> > > - Line 213: I'm still not convinced of the claim that the matrix R is approximately diagonal. No evidence is shown in favour of this approximation but the main theoretical contributions of this work (theorems 4 and 5) depend on it. I would need to see some evidence before I am convinced that this work is any useful.
> > >
> > > - One of the main problems I have with this work is that I do not trust the interpretation of the experimental results given by the authors. Results are weak and inconsistent. It would be much more convincing to test the theory in a more controlled setting. For example, before even talking about how to add momentum to the picture, the authors could show that equation 8 truly improves the prediction of the loss, by just testing the bound provided, with and without the glass term, and show that the glass term significantly improves loss prediction.
> > >
> > > - Line 46: I strongly disagree, Fig.2 only shows difference between "final transformations", which I guess corresponds to the last layer (?) and all other parameters of previous layers. That is somewhat obvious because "final transformations" is not followed by any nonlinearity. There is no dependence between layer index and the exponent of Figure 2, therefore is no evidence for the claim that "parameters in early layers ... produces higher densities of gradient jumps". That claim is unsupported and should be removed.
> > >
> > > - Does equation 8 also assumes that the Hessian is diagonal? Where is this assumption stated and why is it reasonable?
> > >
> > > - Figure 1: What is the "simplified 2D model"?. How do I reproduce these results? There is no explanation. I'm not arguing that explanations should be in the main text, however Appendix should contain all the information necessary to reproduce the results.
> > >
> > > - Line 239: "the increase in loss ... is maximized if a local floor enforces ..." I don't understand what that means. Is equation 7 an average over delta. If so, then why does the result depend on delta?
> > >
> > > - I guess the average in Equation 1 is taken over the distribution of delta vectors, however that is not explained.
> > >
> > > - Line 96: What is the "beta2 schedule"? Is it akin to momentum? All of this should be explained in the Appendix, not the main text.
> > >
> > > - Line 53: "average training loss". Average over what?

---

> > > > ### Author Response · Authors · 2025-11-25
> > > >
> > > > We thank the reviewer again for their follow-up comments. We address each remaining point below and describe the clarifications we have incorporated into the revision.
> > > >
> > > > > Diagonal dominance of R
> > > >
> > > > Using a diagonal approximation follows long-established practice in scalable second-order optimization. Diagonal curvature has been used since Becker & LeCun (1988) and LeCun, Denker & Solla (1990), and is treated as the standard tractable approximation for large models in contemporary surveys such as Bottou, Curtis & Nocedal (2018). Modern second-order optimizers—including AdaHessian (Yao et al., 2021) and Sophia (Liu et al., 2023)—continue to rely on diagonal curvature for the same scalability reasons. Our use of a diagonal Hessian therefore aligns with both the classical and modern literature on practical quasi-Newton methods.
> > > >
> > > > That said, we agree that the justification cannot simply be inherited from diagonal Hessian practice. In fact, the rationale is *stronger* here. The entries of the glass-density matrix arise from activation-boundary crossings, and these crossings are most sensitive to perturbations aligned with the largest pre-activation-gradient coordinates. As shown in Eq. (3), this produces cubic scaling on the diagonal but only quadratic–linear scaling off it, so diagonal dominance is structurally built into the operator. *This analytic asymmetry does not hold for Hessians*, where diagonal approximations are nevertheless considered standard and widely used for scalability.
> > > >
> > > > We have clarified directly in the manuscript why a diagonal approximation is both analytically supported and practically necessary. Equation (3) shows the cubic–vs.–quadratic–linear scaling that concentrates mass on diagonal entries, and we now explicitly note that summing over many units amplifies this effect. We also state that even just storing full off-diagonal structure would require d^2 elements, which is infeasible at modern parameter scales.
> > > >
> > > >
> > > > > Role of Eq. (8) and whether it should be empirically validated
> > > >
> > > > Eq. (8) is not intended as a predictive model of empirical loss behavior; rather, it is the majorized surrogate used in the quasi-Newton derivation. We now state this explicitly. Equation (8) supplies the 3/2-order curvature term in a convex upper model, and Theorem 5 solves the resulting coordinate-wise minimization to obtain the quasi-Newton update. This is now clearly stated to prevent conflation with empirical loss prediction.
> > > >
> > > > > Layer-dependence of exponents in Fig. 2
> > > >
> > > > The revised text no longer refers to “early layers.” Instead, we now state, consistently with Fig. 2, that parameters followed by ReLUs exhibit p < 2 exponents because perturbations cross downstream activation boundaries, whereas parameters not followed by ReLUs (e.g., the final block) recover the Hessian scaling p = 2. This directly resolves the concern.
> > > >
> > > > > Diagonal Hessian assumption in Eq. (8)
> > > >
> > > > We now state directly that Eq. (8) uses a diagonal Hessian approximation, again following standard practice in large-scale quasi-Newton methods. The text explicitly notes that this is the only tractable option at model scale.
> > > >
> > > > > Reproducibility of the simplified 2D model (Fig. 1)
> > > >
> > > > A concise pseudocode description has been added to Appendix B. It explains how boundary anchors are sampled, lightly repelled for uniform spacing, assigned gradient jumps, and intersected along a trajectory to create the gradient and loss envelopes. A pointer to Appendix B has been added to the caption.
> > > >
> > > > > Meaning of the “local floor”
> > > >
> > > > We have clarified that the floor simply converts symmetric, zero-mean gradient-jump fluctuations into a folded-normal expectation, which is required to construct a valid convex majorizer. Without the floor, the expectation would be zero. This short clarification now appears in the text.
> > > >
> > > > > Expectation notation in Eq. (1)
> > > >
> > > > We have changed the notation to E_\delta for clarity.
> > > >
> > > > > Meaning of “β₂ schedule”
> > > >
> > > > We now refer to it explicitly as the β₂ exponential moving average (as in Adam). This avoids ambiguity.
> > > >
> > > > > “Average training loss”
> > > >
> > > > We now specify it as the average over the training batches.
> > > >
> > > >
> > > > We hope these clarifications fully resolve the reviewer’s concerns. We believe the revised text now addresses each point accurately and directly, and improves clarity throughout.

---

### Meta-Review · Area_Chair_SiaH · 2026-01-07

**Summary:**

The reviewers were split about this paper and did not come to a consensus. On one hand they appreciated the motivation of the paper. On the other they had issues with the clarity of the paper. Two reviewers responded to the author feedback, one to thank the authors and the other to describe multiple concerns that had not been addressed by reviewers. The crux of the reviewer’s argument is that the paper is very unclear and needs to be clarified before the paper is accepted. The authors responded again to this reviewer to clarify additional concerns. One of the main problems that reviewer has is that the theory is not tested in a controlled setting. They argue that without such a test the interpretation of the experimental results cannot be trusted. However, the authors do not directly address this concern. They instead argue that a proposed test by the reviewer does not make sense. This large concern still stands. This is combined with the fact that the paper is still very hard to understand even after the changes, e.g., “by folding symmetric gradient-jump effects to obtain a valid convex upper model for the expectation” (this may be due to the use of GPT-5 which is notoriously bad at generating clear technical explanations). If the authors can clarify the paper and develop a better-controlled experimental procedure to test their interpretations, the paper will be much improved. As it is right now, I vote to reject.

**Reviewer Concerns:**

Please see above.

**Reviewer Scores:**

I believe they would have kept their scores or decreased them.

---

### Decision · Program_Chairs · 2026-01-26

Reject